# Application of Quantitative Ultrasonography and Artificial Intelligence for Assessing Severity of Fatty Liver: A Pilot Study

**DOI:** 10.3390/diagnostics14121237

**Published:** 2024-06-12

**Authors:** Hyuksool Kwon, Myeong-Gee Kim, SeokHwan Oh, Youngmin Kim, Guil Jung, Hyeon-Jik Lee, Sang-Yun Kim, Hyeon-Min Bae

**Affiliations:** 1Laboratory of Quantitative Ultrasound Imaging, Seoul National University Bundang Hospital, Seong-nam 13620, Republic of Korea; jinuking3g@snubh.org (H.K.); mgkim@barreleye.co.kr (M.-G.K.); joseph9337@kaist.ac.kr (S.O.); 2Imaging Division, Department of Emergency Medicine, Seoul National University Bundang Hospital, Seong-nam 13620, Republic of Korea; 3Barreleye Inc., 312, Teheran-ro, Gangnam-gu, Seoul 06221, Republic of Korea; 4Electrical Engineering Department, Korea Advanced Institute of Science and Technology, Daejeon 34141, Republic of Korea; youngmin2007@kaist.ac.kr (Y.K.); jgl97123@kaist.ac.kr (G.J.); dlguswlr0811@kaist.ac.kr (H.-J.L.); kmjmksy@kaist.ac.kr (S.-Y.K.)

**Keywords:** non-alcoholic fatty liver disease, artificial intelligence, quantitative ultrasound, attenuation coefficient, non-invasive diagnosis

## Abstract

Non-alcoholic fatty liver disease (NAFLD), prevalent among conditions like obesity and diabetes, is globally significant. Existing ultrasound diagnosis methods, despite their use, often lack accuracy and precision, necessitating innovative solutions like AI. This study aims to validate an AI-enhanced quantitative ultrasound (QUS) algorithm for NAFLD severity assessment and compare its performance with Magnetic Resonance Imaging Proton Density Fat Fraction (MRI-PDFF), a conventional diagnostic tool. A single-center cross-sectional pilot study was conducted. Liver fat content was estimated using an AI-enhanced quantitative ultrasound attenuation coefficient (QUS-AC) of Barreleye Inc. with an AI-based QUS algorithm and two conventional ultrasound techniques, FibroTouch Ultrasound Attenuation Parameter (UAP) and Canon Attenuation Imaging (ATI). The results were compared with MRI-PDFF values. The intraclass correlation coefficient (ICC) was also assessed. Significant correlation was found between the QUS-AC and the MRI-PDFF, reflected by an R value of 0.95. On other hand, ATI and UAP displayed lower correlations with MRI-PDFF, yielding R values of 0.73 and 0.51, respectively. In addition, ICC for QUS-AC was 0.983 for individual observations. On the other hand, the ICCs for ATI and UAP were 0.76 and 0.39, respectively. Our findings suggest that AC with AI-enhanced QUS could serve as a valuable tool for the non-invasive diagnosis of NAFLD.

## 1. Introduction

Non-alcoholic fatty liver disease (NAFLD) has emerged as a significant global health challenge, characterized by excessive fat accumulation in the liver in the absence of substantial alcohol consumption. The rising prevalence of NAFLD is linked to increasing rates of obesity, metabolic syndrome, and type 2 diabetes [1,2,3]. This trend highlights the urgent need for effective early detection methods to prevent the progression of NAFLD and reduce the risk of serious complications such as hepatic cirrhosis and hepatocellular carcinoma [4,5].

Liver biopsy, the current gold standard for diagnosing NAFLD, provides the direct visualization and quantification of lipid accumulation. However, its invasive nature, high cost, and associated risks limit its applicability for routine screening and early detection in large populations [6]. This limitation underscores the necessity for developing non-invasive, accurate, and reliable diagnostic and monitoring tools for NAFLD.

Traditional imaging techniques, such as abdominal ultrasonography (US), have been widely utilized for non-invasive liver disease detection. Nonetheless, their varying sensitivities and specificities in identifying and quantifying hepatic steatosis necessitate the adoption of more sophisticated methods like Quantitative Ultrasonography (QUS) [7,8]. QUS evaluates steatosis through the Ultrasound Attenuation Parameter (UAP), which quantifies liver fat content using dedicated devices that do not assess liver morphology. Despite these advancements, manual settings required by operators can introduce inter-observer variability. Jasper et al. developed an AI-based system for liver fat quantification using MRI images, achieving high accuracy but with significant computational costs [9].

Artificial intelligence (AI) offers a promising solution to enhance the accuracy of QUS by reducing inter-observer variability and autonomously analyzing liver morphology and raw ultrasound data. This paper introduces an AI-enhanced QUS method that utilizes an attenuation coefficient (QUS-AC, Barreleye Inc., Seoul, Republic of Korea) to evaluate NAFLD severity. We aim to compare the efficacy of this AI-based algorithm against traditional UAP and ATI (Attenuation Imaging), an FDA-approved technique implemented in the Aplio i800 US system (Canon Medical Systems, Tochigi, Japan). These measurements will be compared, with MRI-derived Proton Density Fat Fraction (PDFF) serving as the reference standard. Additionally, this pilot study investigates the reliability of our algorithm by examining inter-observer variability [10].

This study advances the field by demonstrating the efficacy of an AI-enhanced QUS algorithm for assessing NAFLD severity. It offers a non-invasive, accurate, and cost-effective alternative to existing diagnostic methods. The integration of AI addresses inter-observer variability, providing consistent and reliable measurements.

The remainder of this paper includes a description of the materials and methods used in this study, including participant recruitment, imaging techniques, and AI algorithm development. It then presents the results, discusses the findings in the context of the existing literature, and concludes with key outcomes and future research directions.

## 2. Materials and Methods

### 2.1. Study Design and Participants

This study was a single-center, cross-sectional pilot study conducted from September to October 2022 (Figure 1). We recruited adult participants (age >18 years) who were scheduled for a routine health check-up and were suspected of having fatty liver based on their clinical history, physical examination, and routine blood tests. We excluded participants with a history of alcohol abuse, viral hepatitis, or other known liver diseases.

Ultrasound examinations were conducted by radiologists with a decade or more of experience. All radiologists were trained to perform the examination and analysis using the same standard protocol to minimize operator variation. The patients were supine with their right arm abducted in a quiet, dimly lit room to minimize respiratory motion impact. Liver scans were performed in longitudinal and transverse planes, with gain settings and focus adjusted for optimal image quality. Five measurements were taken from the right liver lobe through intercostal spaces. The ultrasound examinations were performed on the same day as the MRI scans to ensure consistency in liver fat measurements.

For each patient, after the B-mode ultrasound examination, the same radiologist performed additional ultrasound examinations on the same day. Each session was performed using three different ultrasound examination platforms: QUS-AC (dB/cm/MHz) using Vantage 64LE (Verasonics, Inc., Kirkland, WA, USA) ultrasound system (Figure 2B), ATI (dB/cm/MHz) using Aplio i900 (Canon Medical System, Tochigi, Japan) (Figure 2C), and UAP (dB/m) using FT100 (FibroTouch, Wuxi, Jiangsu, China) (Figure 2D).

The results of each ultrasound examination were derived as the average of five independent measurements. For the inter-observer reliability analysis, two independent radiologists, blinded to the MRI results and the measurements of the other radiologist, analyzed the output values derived from the ultrasound examination.

Written informed consent was obtained from all participants. The study protocol was approved by the institutional review board (KH2020-152).

### 2.2. Functional MRI Acquisition

MRI examinations were performed using a 3.0-Tesla scanner (Magnetom Verio 3T, Siemens, Munich, Germany). Participants were asked to fast for at least six hours before the MRI scan to minimize the effects of food intake on liver fat measurements.

A multiparametric MRI protocol was used, which included T1-weighted imaging, T2-weighted imaging, and a Proton Density Fat Fraction (PDFF) sequence for the quantification of liver fat. The PDFF sequence was a multi-echo gradient echo sequence acquired in a single breath-hold, with six echoes used to separate water and fat signals.

The PDFF images were reconstructed offline using a complex-based water–fat separation algorithm. This algorithm estimates the PDFF, which measures the proportion of the fat signal relative to the total signal (water and fat) in each voxel. The PDFF maps were used to quantify liver fat and are expressed as a percentage (%) (Figure 2A).

Regions of interest (ROIs) for PDFF measurements were manually drawn on the PDFF maps by a radiologist, who was blinded to the results of the ultrasound examinations. The ROIs were placed in the approximate anatomical locations for the ultrasound examinations, avoiding visible vessels, bile ducts, and focal liver lesions. The mean PDFF value within the ROI was used as a quantitative measure of liver fat content.

The entire MRI procedure, including patient preparation, scan acquisition, and image analysis, took approximately 20–30 min per patient. All MRI scans were performed and analyzed according to the same standard protocol to ensure consistency and reliability. In this study, the PDFF cutoff values for diagnosing steatosis grades of ≥1, ≥2, and 3 were determined to be 5%, 16.3%, and 21.6%, respectively.

### 2.3. AI-Based QUS Algorithm Development and Validation

In this paper, we propose an AI-based quantitative ultrasound algorithm that estimates the AC of liver parenchyma from the captured ultrasound signals. To mitigate the decrease in accuracy caused by ultrasound noise reflected from tissues other than the liver and reduce measurement variation among users, the proposed AI-based algorithm adaptively selects the parenchymal area of the liver from B-mode images and derives the AC values using only ultrasound signals reflected from the liver parenchyma (Figure 2B).

### 2.4. Attenuation Imaging

ATI, an FDA-approved technique implemented in the Aplio i800 US system (Canon Medical Systems, Tochigi, Japan), quantifies ultrasound attenuation in tissue using real-time color mapping over a large sample area. The measurement box has a fixed size of 30 × 30 mm when positioned in the image center. Operators can freely move both the region of interest (ROI) and the measurement box using the trackball of the US system to select the measurement area. The entire ATI examination took approximately 5 min per patient (Figure 2C).

### 2.5. UAP and Stiffness Assessment

UAP was obtained using the FT100 (FibroTouch, Wuxi, Jiangu, China). The 3.5 MHz M probe was used. The FibroTouch estimates both liver stiffness in kiloPascals (kPa) and liver attenuation coefficient in decibels/meters (dB/m) (Figure 2D).

#### 2.5.1. Deep Neural Network Architecture

The time-varying intensity of the received pulse-echo data imposes the biomechanical properties including the AC of target tissues. The proposed network extracts the AC value of the liver parenchyma from the pulse-echo envelope-detected data, E_n, obtained using five-angle planewave patterns and the corresponding B-mode image. As shown in Figure 3, the network includes (1) individual encoders extracting the time-varying intensity information of the envelope-detected pulse-echo data through convolutional computation, (2) an adaptive normalization layer utilizing B-mode images to determine the representative feature map from aggregated encoded data, and (3) consecutive convolutional layers extracting the AC value from B-mode-guided normalized features (Figure 3).

#### 2.5.2. Encoder

Each envelope-detected data item E_(1:5) is encoded through individual convolutional computations to extract intensity variation over time. Each output of an individual encoding path is concatenated channel-wise to mitigate the information from different beam patterns. A residual method is applied in the encoder to address the vanishing/exploding gradient problem [11]. 

#### 2.5.3. B-Mode-Guided Adaptive Denormalization

In order to concentrate selectively on the features containing the AC information of the liver parenchyma in the encoded feature map, the B-mode image containing the morphological information of the liver parenchyma is utilized. The morphology visible in the B-mode image is reflected in the time-information of the envelope-detected data. Inspired by the adaptive instance normalization (18), the encoded profile is adaptively normalized in the time domain and subsequently scaled and shifted using B-mode-dependent parameter vectors γ_t and β_t. The B-mode-guided adaptive denormalization (BGN) is given by
(1)$$X_{t}=\gamma_{t}\times \frac{x_{t}-\mu }{\sigma }+\beta_{t}$$
where x_t is the encoded data of the t-th time, X_t is the denormalized feature for the t-th time, μ and σ are the mean and standard deviation of x_t, respectively. The B-mode-dependent parameter vectors γ_t and β_t are extracted by processing the B-mode image through 4 common convolutional layers and 4 individual fully connected layers. The B-mode-guided normalized feature map is computed through a series of convolutional computations with batch normalization to manage the internal covariate shift problem [12]. Lastly, the AC value of the liver is inferred by using fully connected layers.

#### 2.5.4. Synthetic Dataset

It is highly challenging to acquire ultrasound data according to the quantified liver attenuation coefficients in a clinical setting. To overcome this issue, we constructed an environment where ultrasound data corresponding to desired liver attenuation coefficients could be obtained using an ultrasound computer simulator [10]. The ultrasound simulator was based on the k-wave simulation tool in MATLAB [11]. The simulation environment was modeled based on Vantage 64LE ultrasound system (Verasonics Inc., Kirkland, WA, USA) with a convex array probe (Humanscan, Siheung-si, Republic of Korea). A total of 64 channels were used to transmit and receive five different plane waves with the angles of −7.2°, −3.6°, 0°, 3.6°, and 7.2°.

The simulation phantom representing liver parenchyma and abdominal blood vessels was created by placing one ellipse with a radius of 50 mm to 80 mm and less than 10 objects with a radius of 3 mm to 50 mm in arbitrary locations, respectively. In addition, 6 to 10 layers were placed within a 30 mm depth to represent muscle, subcutaneous fat, and skin. (Figure 4) The wave propagation properties of the tissue were modeled based on [10] (Table 1). In order to model the deformation of the superficial tissues due to the shape of the convex array probe, the axial grid was warped accordingly to match the curvature of the probe. Less than 10 scatterers with densities ranging from 800 kg/m^3^ to 1200 kg/m^3^ were uniformly placed in the unit area (wavelength by wavelength area).

#### 2.5.5. Implementation Details

Out of the total 10,000 ultrasound image datasets, 8000 were used for training, 1000 for validation, and 1000 for testing. Care was taken to ensure that images from the same patient did not end up in more than one category to prevent data leakage. Based on supervised learning, the training objective of the network was to minimize the L1 loss between the estimated AC values and the ground truth AC, which represents the AC values obtained from the MRI-PDFF as the reference standard. L2 regularization was applied to enhance the convergence of the network [11]. The network was trained using the Adam optimization algorithm [12] with an initial learning rate of 5 × 10^−6^, and training was stopped when the loss on the validation set converged. The network was trained and tested using PyTorch, accelerated with an NVIDIA RTX3090 GPU. The performance of the proposed method was evaluated by using the mean normalized absolute error (MNAE) in the test dataset. The proposed algorithm achieved 6.49% in MNAE, which corresponded to an improvement of 41% in MNAE against the algorithm that did not use B-mode images.

### 2.6. Primary Endpoint

The primary endpoint of our study was to establish the correlation between the AC values estimated using our AI-based QUS algorithm and the MRI-PDFF values. In addition to comparing these values with MRI results, we conducted a similar analysis using two ultrasound devices (Canon ATI and FibroScan UAP (Echosens, Hong Kong, China)). The correlation between the fat contents estimated using these conventional devices and the MRI-PDFF was also examined, allowing us to compare the effectiveness and accuracy of our newly developed AI-based QUS algorithm with these existing ultrasound technologies. We also compared the discriminatory power and accuracy for the grade of steatosis.

### 2.7. Secondary Endpoint

The secondary endpoint was the inter-observer reliability of our algorithm. Two independent observers, who were blinded to the MRI results, applied the algorithm to the ultrasound data. Inter-observer reliability was assessed using the intraclass correlation coefficient (ICC).

### 2.8. Statistical Analyses

All statistical analyses were performed using R software package version 4.3.3. Continuous variables were presented as mean ± standard deviation (SD), while categorical variables were presented as frequencies and percentages. For the primary endpoint, we used Spearman’s correlation coefficient to assess the relationship between the AC values estimated via our algorithm and the PDFF values obtained from the MRI scans. Additionally, we assessed the diagnostic accuracy of our algorithm by calculating the Area Under the Receiver Operating Characteristic (AUROC) curve, providing a comprehensive evaluation of the algorithm’s performance across various thresholds. A *p*-value less than 0.05 was considered statistically significant. For the secondary endpoint, inter-observer reliability was evaluated by comparing the AC values estimated by two independent observers blinded to the MRI results. Differences in AC values were examined to assess if they were related to variations in region-of-interest selection among the observers. We calculated the ICC to assess inter-observer reliability, using a mean-rating (k = 2), absolute-agreement, 2-way random-effects model. The ICC estimates and their 95% confidence intervals (CIs) were calculated, with ICC values categorizing reliability as follows: less than 0.5 indicated poor reliability, between 0.5 and 0.75 indicated moderate reliability, between 0.75 and 0.9 indicated good reliability, and greater than 0.9 indicated excellent reliability.

## 3. Results

### 3.1. Participants Characteristics

This study comprised 35 participants (Table 2), with a mean age of 52.3 ± 14.5 years. Gender distribution was 60% male (*n* = 21) and 40% female (*n* = 14). The mean Body Mass Index was 28.7 ± 5.1 kg/m^2^. Among the cohort, the prevalences of diabetes, hypertension, and hyperlipidemia were 29% (*n* = 10), 49% (*n* = 17), and 40% (*n* = 14), respectively. Regarding alcohol consumption, 29% (*n* = 10) of the participants reported occasional intake, whereas 71% (*n* = 25) reported no alcohol consumption. The mean Aspartate Aminotransferase level was 32.6 ± 11.2 U/L, and the mean Alanine Aminotransferase level was 45.4 ± 18.3 U/L. The mean MRI-PDFF was 11.16 ± 7.24%, the mean UAP was 262.54 ± 46.66 dB/m, the mean ATI value was 0.71 ± 0.09 dB/cm/MHz, and the mean QUS-AC value was 0.44 ± 0.09 dB/cm/MHz.

### 3.2. Primary Endpoint

A significant correlation was observed between the QUS-AC and the MRI-PDFF in the linear regression analysis, as demonstrated by an r coefficient of determination of 0.95. The R-squared value for QUS-AC was 83.86 (*p* < 0.001), and the 95% CI ranged from 77.15 to 90.58 (Figure 5A). The linear regression analysis between the MRI-PDFF and ATI resulted in an r coefficient of determination of 0.73 and an R-squared value of 65.37 (95% CI: 45.64 to 85.11) (Figure 5B). UAP and MRI-PDFF demonstrated the weakest correlation (r coefficient value: 0.51; R-squared value: 1.31; 95% CI: −0.19 to 2.82) (Figure 5C). 

For QUS-AC, the cutoff values are 0.36 dB/cm/MHz for Grade 1, 0.46 dB/cm/MHz for Grade 2, and 0.53 dB/cm/MHz for Grade 3. In the case of ATI, the thresholds are set at 0.63 dB/cm/MHz for Grade 1, 0.71 dB/cm/MHz for Grade 2, and 0.81 dB/cm/MHz for Grade 3. Additionally, UAP measurements indicate cutoffs of 222 dB/m for Grade 1, 280 dB/m for Grade 2, and 281 dB/m for Grade 3 (Figure 6). The diagnostic accuracy, represented by AUROC, ranges from 0.93 to 0.99 across these methods and grades, illustrating high reliability in assessing fatty liver severity (Table 3).

### 3.3. Secondary Endpoint

The ICC was calculated to assess the reliability of observations and the mean reliability of the proposed methods for assessing NAFLD (Table 4). This analysis was based on a total of 35 subjects, with two raters for each. The ICC for individual observations of AC was found to be 0.98, with a 95% CI ranging from 0.97 to 0.99. The ICC for the mean reliability of AC was observed to be 0.99, with a 95% confidence interval ranging from 0.98 to 0.99. For SWE, the ICC for individual observations was 0.3, with a 95% CI ranging from 0.22 to 0.38, while the ICC for the mean reliability was 0.39, with a 95% CI ranging from 0.28 to 0.49. Regarding ATI, the ICC for individual observations was 0.7, with a 95% CI ranging from 0.61 to 0.79. and the ICC for the mean reliability was 0.76, with a 95% CI ranging from 0.69 to 0.82.

## 4. Discussion

In this pilot study, we evaluated the effectiveness of an AI-enhanced quantitative ultrasound (QUS) algorithm, focusing on the attenuation coefficient (AC) as a metric for assessing the severity of non-alcoholic fatty liver disease (NAFLD). Our findings indicate a substantial and robust correlation between the AC values derived from AI-QUS and those from the MRI Proton Density Fat Fraction (MRI-PDFF), a well-established method in liver diagnostics. Our findings align with those of Jasper et al., who reported high accuracy in AI-based liver fat quantification using MRI. However, our study extends this work by demonstrating the feasibility of using AI-enhanced QUS, which offers advantages in terms of cost and accessibility [9]. The AI-enhanced QUS algorithm not only matches but potentially exceeds the predictive capabilities of current non-invasive diagnostic tools such as the Attenuation Imaging Technique (ATI) and Ultrasound Attenuation Parameter (UAP), which demonstrated significantly lower correlations in our study.

The introduction of AI into QUS represents a pivotal shift in ultrasound diagnostics. Traditional QUS, while beneficial, is limited by its sensitivity to operator skill and the subjective interpretation of ultrasound data. By standardizing measurements, our AI algorithm addresses these limitations, reducing the potential for human error and providing a more objective and reproducible assessment of liver steatosis. This is crucial, considering the nuanced nature of NAFLD, where early and accurate detection is paramount for managing the disease effectively. By automating the extraction and analysis of ultrasound data, the AI-enhanced QUS algorithm reduces inter-observer variability—a common challenge in ultrasound diagnostics. This consistency is supported by the high inter-observer reliability scores (ICC = 0.983) observed in our study, suggesting that AI-enhanced QUS provides consistent and reliable measurements across different clinicians.

The potential for the AI-QUS algorithm to be integrated into routine clinical practice is significant, especially in settings where NAFLD is prevalent but underdiagnosed. The non-invasive nature of AI-enhanced QUS, combined with its demonstrated accuracy, makes it a valuable tool for the early detection and monitoring of NAFLD. It offers a safer alternative to liver biopsies, which are invasive and carry risks of complications. Furthermore, the use of AI-QUS could lead to substantial cost savings for healthcare systems, as it could reduce the need for more expensive imaging tests and invasive procedures.

However, the results of our pilot study must be contextualized within its limitations. The small sample size and single-center design may impact the external validity and generalizability of the findings. The homogeneity of the study population also raises questions about the applicability of AI-QUS across diverse populations, who may present with varying stages of NAFLD and associated metabolic conditions. Additionally, as with any AI-driven technology, the efficacy of AI-QUS is contingent on the quality and diversity of the data used for training the models. Biases in training data can lead to skewed assessments and potentially misinform clinical decision-making.

Looking forward, expanding the capabilities of AI-QUS to other hepatic conditions and potentially other organ systems could broaden the impact of this technology. Research could also explore integrating AI-enhanced QUS with other diagnostic modalities to create a comprehensive diagnostic toolkit for liver diseases. Moreover, further development in AI algorithms could focus on improving their adaptability to new data, enhancing their ability to learn from ongoing clinical use and feedback.

## 5. Conclusions

The AI-enhanced QUS algorithm represents a significant step forward in the non-invasive assessment of liver steatosis and holds promise as a potential standard in liver diagnostics across various conditions. By continuing to refine and validate this technology, we can hope to see it become a cornerstone of clinical practice, offering a safer, more accurate, and cost-effective alternative to traditional diagnostic methods. Our results indicate an ICC of 0.983 for QUS-AC, demonstrating high reliability. The AI algorithm showed a strong correlation with MRI-PDFF values (R^2^ = 0.95), outperforming traditional ultrasound methods like ATI (R^2^ = 0.58) and UAP (R^2^ = 0.09). These findings suggest that AI-enhanced QUS could serve as a valuable tool for the early detection and monitoring of NAFLD, offering a non-invasive, accurate, and cost-effective alternative to liver biopsies and other imaging techniques. Future research should focus on expanding the capabilities of AI-QUS to other hepatic conditions and integrating it with other diagnostic modalities to create a comprehensive diagnostic toolkit for liver diseases.

## Figures and Tables

**Figure 1 diagnostics-14-01237-f001:**
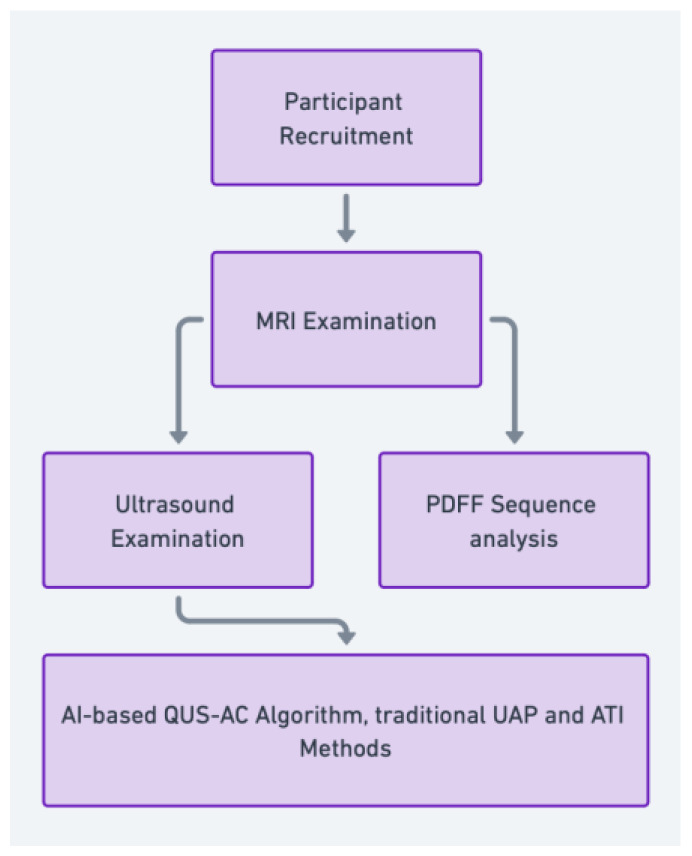
Participants were recruited and underwent MRI examinations, including Proton Density Fat Fraction (PDFF) sequence analysis, and ultrasound examinations using the quantitative ultrasound attenuation coefficient (QUS-AC), Ultrasound Attenuation Parameter (UAP), and Attenuation Imaging (ATI) techniques. The AI-based QUS-AC algorithm and traditional methods were then used to evaluate liver fat content.

**Figure 2 diagnostics-14-01237-f002:**
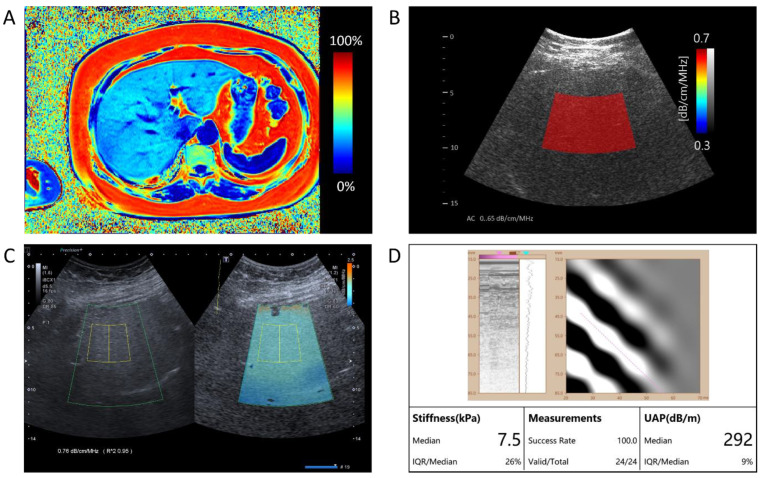
A comprehensive assessment of hepatic fat content in a single patient. (**A**) The representation of Magnetic Resonance Imaging-Proton Density Fat Fraction (MRI-PDFF) presents a measured value of 31.59%, quantifying the level of hepatic steatosis. (**B**) The AI-enhanced quantitative ultrasound attenuation coefficient (QUS-AC), quantified as 0.65 dB/cm/MHz, demonstrates the AI-based quantitative ultrasound assessment of fatty liver. (**C**) The Canon ATI returns an attenuation value of 0.76 dB/cm/MHz. (**D**) The FibroTouch UAP returns an attenuation value of 292 dB/m.

**Figure 3 diagnostics-14-01237-f003:**
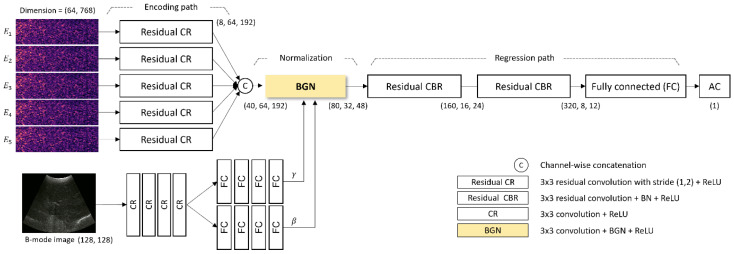
The architecture of the proposed deep neural network for estimating the AC value from the captured ultrasound signals. BGN: B-mode-guided normalization; CBR: Convolutional block.

**Figure 4 diagnostics-14-01237-f004:**
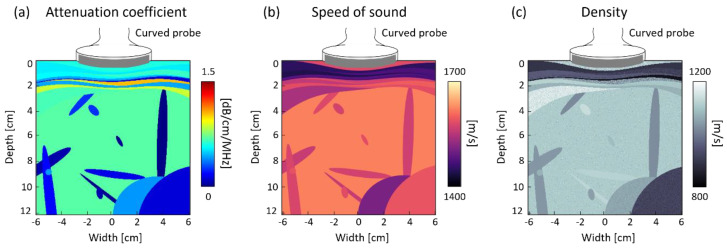
The characteristic of the simulation phantom.

**Figure 5 diagnostics-14-01237-f005:**
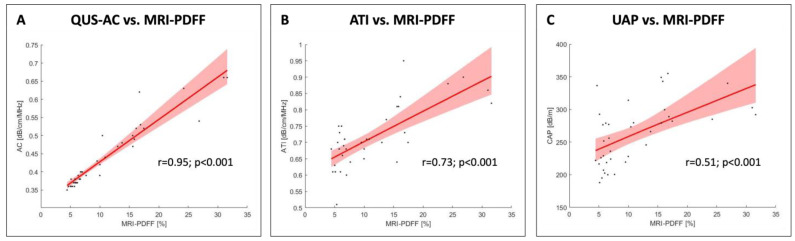
The linear regression analysis revealed a significant correlation between AC and MRI-PDFF with an r value of 0.95 and an R-squared value of 83.86 (95% CI: 77.15 to 90.58). A weaker correlation was found between ATI and MRI-PDFF (r value: 0.73; R-squared value: 65.37; 95% CI: 45.64 to 85.11). UAP and MRI-PDFF demonstrated the weakest correlation (r value: 0.51; R-squared value 1.31; 95% CI: −0.19 to 2.82).

**Figure 6 diagnostics-14-01237-f006:**
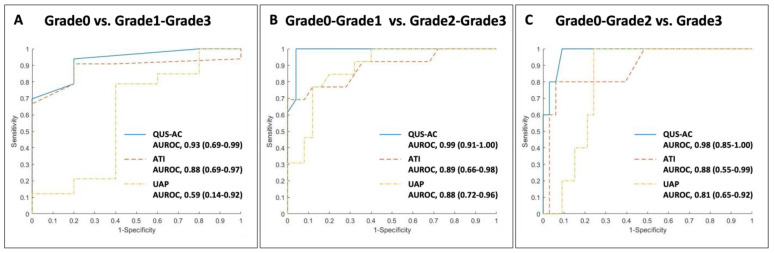
Receiver Operating Characteristic curves comparing AI-enhanced quantitative ultrasound attenuation coefficient (QUS-AC, Barreleye Inc., Seoul, Republic of Korea), ATI in the Aplio i800 US system (Canon Medical Systems, Tochigi, Japan), and traditional UAP.

**Table 1 diagnostics-14-01237-t001:** The acoustic properties of the synthetic phantom.

Organ	SoS [m/s]	AC [dB/cm/MHz]	Density [kg/m^3^]
Liver parenchyma	1500–1600	0–1.0	1050–1150
Blood vessel	1500–1600	0–0.5	1050–1150
Muscle	1500–1650	0–1.5	1000–1200
Fat	1400–1500	0–0.5	800–950
Skin	1500–1700	0–1.0	1100–1150
Other organs in abdomen	1450–1650	0–1.5	850–1150

SoS: Speed of Sound, AC: Attenuation Coefficient.

**Table 2 diagnostics-14-01237-t002:** Baseline demographic and clinical characteristics of the study population.

Characteristic	Value
Total number of participants	35
Age, mean (SD) years	52.3 (14.5)
Male, No. (%)	21 (60%)
Female, No. (%)	14 (40%)
BMI, mean (SD) kg/m^2^	28.7 (5.1)
Diabetes, No. (%)	10 (29%)
Hypertension, No. (%)	17 (49%)
Hyperlipidemia, No. (%)	14 (40%)
MRI-PDFF (%), mean (SD)	11.16 (7.24)
UAP (dB/m), mean (SD)	262.54 (46.66)
ATI (dB/cm/MHz), mean (SD)	0.71 (0.09)
AC (dB/cm), mean (SD)	0.44 (0.09)

AC: attenuation coefficient; ATI: an FDA-approved technique implemented in the Aplio i800 US system (Canon Medical Systems, Tochigi, Japan); UAP: Ultrasound Attenuation Parameter; MRI-PDFF: Magnetic Resonance Imaging Proton Density Fat Fraction; SD: standard deviation; BMI: Body Mass Index.

**Table 3 diagnostics-14-01237-t003:** Diagnostic accuracy of AI-enhanced quantitative ultrasound attenuation coefficient (QUS-AC), ATI in the Aplio i800 US system (Canon Medical Systems, Japan), and traditional UAP.

Parameter	Method	Grade 0 vs. Grade 1–Grade 3	Grade 0–Grade 1 vs. Grade 2–Grade 3	Grade 0–Grade 2 vs. Grade 3
Cutoff in dB/cm/MHz	QUS – AC	0.36	0.46	0.53
Cutoff in dB/cm/MHz	ATI	0.63	0.71	0.81
Cutoff in dB/m	UAP	222	280	281
AUROC	QUS–AC	0.93 (0.69–0.99)	0.99 (0.91–1.00)	0.98 (0.85–1.00)
	ATI	0.88 (0.69–0.97)	0.89 (0.66–0.98)	0.88 (0.55–0.99)
	UAP	0.59 (0.14–0.92)	0.88 (0.72–0.96)	0.81 (0.65–0.92)
Sensitivity	QUS–AC	0.94	1.00	0.80
	ATI	0.91	0.77	0.80
	UAP	0.79	0.77	1.00
Specificity	QUS–AC	0.80	0.96	0.97
	ATI	0.80	0.88	0.94
	UAP	0.60	0.88	0.76
Positive predictive value	QUS–AC	0.97	0.93	0.80
	ATI	0.97	0.77	0.67
	UAP	0.93	0.77	0.38
Negative predictive value	QUS–AC	0.67	1.00	0.97
	ATI	0.57	0.88	0.97
	UAP	0.30	0.88	1.00
Positive likelihood ratio	QUS–AC	4.70	25.00	24.40
	ATI	4.55	6.41	13.20
	UAP	1.97	6.41	4.13
Negative likelihood ratio	QUS–AC	0.08	0	0.21
	ATI	0.11	0.26	0.21
	UAP	0.35	0.26	0

**Table 4 diagnostics-14-01237-t004:** Intraclass correlation coefficients (ICCs) of various quantitative methods for NAFLD assessment.

Quantitative Methods for NALFD Assessment	Number of Subjects	Number of Raters	The ICC for Individual Observation	95% CI	The ICC for the Mean Reliability	95% CI
QUS-AC	38	2	0.98	0.966–0.992	0.992	0.983–0.996
ATI	38	2	0.70	0.610–0.790	0.760	0.690–0.820

ICC: intraclass correlation coefficient; CI: confidence interval; QUS-AC: AI-enhanced quantitative ultrasound attenuation coefficient; ATI: Attenuation Imaging using Aplio i900 (Canon Medical System).

## Data Availability

The data presented in this study are available on request from the corresponding author. The data are not publicly available due to Barreleye Inc. having intellectual property rights.

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
