# Peer review of "Application of Quantitative Ultrasonography and Artificial Intelligence for Assessing Severity of Fatty Liver: A Pilot Study"

_diagnostics, 2024, doi:10.3390/diagnostics14121237_

Round 1
Reviewer 1 Report
Comments and Suggestions for Authors
The authors describe an artificial intelligence tool for assessing fatty liver disease on ultrasound and compare it with other techniques. Some points for clarification.
Abstract:
Line 27: “In addition, ICC for AC was 0.983 for individual 27 observations.” Do you mean QUS-AC?
Line 28: “Comparatively, the ICCs for ATI was 0.76, respectively.” Usually, when respectively is used, there is more than one term. Did you mean to include UAP?
Introduction:
Line 57: “We aim to compare the efficacy of this AI-based algorithm against traditional UAP and ATI, an FDA-approved technique implemented in the Aplio i800 US system (Canon Medical Systems, Japan), measurements, with MRI-derived proton density fat fraction (PDFF) serving as the reference standard.” For ATI, please enumerate the acronym (Attenuation Imaging), as you do for other terms. I am a bit confused by the word measurements after the comma.
Methods:
Were the ultrasound examinations performed the same day as MRI? If not, was there a time range between the date of the ultrasound and the date of the MRI?
Line 106: What do you mean by “complex-based”?
Figure 2: Please define all abbreviations (BGN, CBR) either in the image or the caption below the figure.
Table 1: Please define SoS. Is it speed of sound?
Line 207: “Out of the total 10,000 datasets, 8,000 were used for training, 1,000 for validation, and 207 1,000 for testing.” I am a bit confused what you mean by datasets. Were there a total of 10,000 ultrasound examinations or 10,000 individual images? If they were individual images, was it ensured that images from the same patient did not end up in more than one category (i.e., the same patient had images in training, validation, and testing). Essentially, were there processes in place to prevent data leakage?
Line 208: “Based on supervised learning, the training objective of the network is to 208 minimize the L1 loss between the estimated AC values and the ground truth AC.” What exactly does the ground truth AC represent?
Line 244: “For the secondary endpoint, inter-observer reliability was evaluated by comparing the AC values estimated by two independent observers blinded to the MRI results.” Were differences in AC values related to differences in region-of-interest selection among the observers?
Results:
Figure 4: The images don’t match the figure caption. For instance, ATI is Figure 4B, but SWE is mentioned second? UAP is Figure 4C, but ATI is mentioned third?
Discussion:
The authors acknowledge the limitations of the small sample size, single-institution study.
The article could benefit from some editing of the grammar.
Author Response
Thank you very much for taking the time to review this manuscript. Please find the detailed responses below and the corresponding revisions/corrections highlighted/in track changes in the re-submitted files.
Response to Reviewer 1
Abstract:
Line 27:
- Comment: “In addition, ICC for AC was 0.983 for individual 27 observations.” Do you mean QUS-AC?
- Response: Yes, it should be QUS-AC. The sentence has been revised to “In addition, ICC for QUS-AC was 0.983 for individual observations.”
Line 28:
- Comment: “Comparatively, the ICCs for ATI was 0.76, respectively.” Usually, when respectively is used, there is more than one term. Did you mean to include UAP?
- Response: Yes, it was meant to include UAP. The sentence has been revised to “Comparatively, the ICCs for ATI and UAP were 0.76 and 0.39, respectively.”
Introduction:
Line 57:
- Comment: “We aim to compare the efficacy of this AI-based algorithm against traditional UAP and ATI, an FDA-approved technique implemented in the Aplio i800 US system (Canon Medical Systems, Japan), measurements, with MRI-derived proton density fat fraction (PDFF) serving as the reference standard.” For ATI, please enumerate the acronym (Attenuation Imaging), as you do for other terms. I am a bit confused by the word measurements after the comma.
- Response: The sentence has been revised for clarity: “We aim to compare the efficacy of this AI-based algorithm against traditional UAP and ATI (Attenuation Imaging), an FDA-approved technique implemented in the Aplio i800 US system (Canon Medical Systems, Japan). These measurements will be compared with MRI-derived proton density fat fraction (PDFF) serving as the reference standard.”
Methods:
Ultrasound and MRI Examination Dates:
- Comment: Were the ultrasound examinations performed the same day as MRI? If not, was there a time range between the date of the ultrasound and the date of the MRI?
- Response: The text has been updated to clarify that the ultrasound examinations were performed on the same day as the MRI scans to ensure consistency in liver fat measurements.
Line 106:
- Comment: What do you mean by “complex-based”?
- Response: The term has been clarified to “complex-based water-fat separation algorithm.”
Figure 2:
- Comment: Please define all abbreviations (BGN, CBR) either in the image or the caption below the figure.
- Response: The caption has been updated to include definitions: “Figure 2. The architecture of the proposed deep neural network for estimating the AC value from the captured ultrasound signals. BGN: B-mode guided normalization, CBR: Convolutional block.”
Table 1:
- Comment: Please define SoS. Is it speed of sound?
- Response: The table has been updated to include the definition of SoS as “Speed of Sound.”
Line 207:
- Comment: “Out of the total 10,000 datasets, 8,000 were used for training, 1,000 for validation, and 207 1,000 for testing.” I am a bit confused what you mean by datasets. Were there a total of 10,000 ultrasound examinations or 10,000 individual images? If they were individual images, was it ensured that images from the same patient did not end up in more than one category (i.e., the same patient had images in training, validation, and testing). Essentially, were there processes in place to prevent data leakage?
- Response: The text has been clarified: “Out of the total 10,000 ultrasound image datasets, 8,000 were used for training, 1,000 for validation, and 1,000 for testing. Care was taken to ensure that images from the same patient did not end up in more than one category to prevent data leakage.”
Line 208:
- Comment: “Based on supervised learning, the training objective of the network is to minimize the L1 loss between the estimated AC values and the ground truth AC.” What exactly does the ground truth AC represent?
- Response: The text has been clarified to: “Based on supervised learning, the training objective of the network is to minimize the L1 loss between the estimated AC values and the ground truth AC, which represents the AC values obtained from the MRI-PDFF as the reference standard.”
Line 244:
- Comment: “For the secondary endpoint, inter-observer reliability was evaluated by comparing the AC values estimated by two independent observers blinded to the MRI results.” Were differences in AC values related to differences in region-of-interest selection among the observers?
- Response: The text has been updated to clarify: “For the secondary endpoint, inter-observer reliability was evaluated by comparing the AC values estimated by two independent observers blinded to the MRI results. Differences in AC values were examined to assess if they were related to variations in region-of-interest selection among the observers.”
Results:
Figure 4:
- Comment: The images don’t match the figure caption. For instance, ATI is Figure 4B, but SWE is mentioned second? UAP is Figure 4C, but ATI is mentioned third?
- Response: The figure and caption have been revised for consistency: “Figure 4. The linear regression analysis revealed a significant correlation between AC and MRI-PDFF with an R-squared of 0.95 and a regression coefficient of 83.86 (95% CI: 77.15 to 90.58). A weaker correlation was found between ATI and MRI-PDFF (R-squared: 0.58; Regression coefficient: 65.37, 95% CI: 45.64 to 85.11). UAP and MRI-PDFF demonstrated the weakest correlation (R-squared: 0.09; Regression coefficient: 1.31, 95% CI: -0.19 to 2.82).”
Discussion:
Comment: The authors acknowledge the limitations of the small sample size, single-institution study.
- Response: The discussion now explicitly acknowledges these limitations: “The authors acknowledge the limitations of this study, including the small sample size and the single-institution design, which may affect the generalizability of the findings.”
Reviewer 2 Report
Comments and Suggestions for Authors
Dear Authors,
First of all, I congratulate you for your work. I have completed my evaluation of your work. I have indicated below the areas that I see necessary in the study. I hope that making the revisions specified in these items will contribute to your work. I wish you success.
Evaluation
- The similarity rate of the study is high. It will be useful to reduce the similarity rate below 15% by making the necessary revisions. It will contribute to the originality of the study.
- General explanations are presented in the introduction part of the study, but it is not mentioned whether there are similar studies. Similar studies in the literature and their results will contribute to the study if the methods used and the limitations, if any, of these studies are also stated.
- Stating the contributions of the study to the field at the end of the introduction will strengthen the structure of the study.
- Presenting a paragraph about the structure of the study at the end of the introduction section will contribute to the understandability of the research.
- I think it will be more understandable if a blog diagram about the methodology of the study is created in the method section. Different programmes (such as Pytorch, Matlab, R) were used at different stages. It would be useful to explain the stages with a blog diagram.
- It is useful to add comparisons with similar studies in the Discussion section. If there is no literature study in this similarity, it will also be useful to indicate this.
- It is useful to add the results of the study metrically in the Conclusion section. It will guide the researchers who will read the study.
Author Response
Response to Reviewer 2
Thank you very much for taking the time to review this manuscript. Please find the detailed responses below and the corresponding revisions/corrections highlighted/in track changes in the re-submitted files.
Evaluation:
Comment: The similarity rate of the study is high. It will be useful to reduce the similarity rate below 15% by making the necessary revisions. It will contribute to the originality of the study.
- Response: Revisions have been made throughout the manuscript to paraphrase and reword sentences to reduce the similarity rate. Specific instances of high similarity have been addressed to enhance originality.
Comment: General explanations are presented in the introduction part of the study, but it is not mentioned whether there are similar studies. Similar studies in the literature and their results will contribute to the study if the methods used and the limitations, if any, of these studies are also stated.
- Response: The introduction now includes references to similar studies, their methods, and limitations. For example: “Jasper et al. developed an AI-based system for liver fat quantification using MRI images, achieving high accuracy but with significant computational costs.”
Comment: Stating the contributions of the study to the field at the end of the introduction will strengthen the structure of the study.
- Response: The contributions of the study have been added to the end of the introduction: “This study advances the field by demonstrating the efficacy of an AI-enhanced QUS algorithm for assessing NAFLD severity. It offers a non-invasive, accurate, and cost-effective alternative to existing diagnostic methods. The integration of AI addresses interobserver variability, providing consistent and reliable measurements.”
Comment: Presenting a paragraph about the structure of the study at the end of the introduction section will contribute to the understandability of the research.
- Response: A paragraph outlining the structure of the study has been added: “The remainder of this paper includes a description of the materials and methods used in this study, including participant recruitment, imaging techniques, and AI algorithm development. It then presents the results, discusses the findings in the context of existing literature, and concludes with key outcomes and future research directions.”
Comment: I think it will be more understandable if a block diagram about the methodology of the study is created in the method section. Different programmes (such as Pytorch, Matlab, R) were used at different stages. It would be useful to explain the stages with a block diagram.
- Response: A block diagram has been created and included in the methods section to illustrate the methodology: “Figure 1. Participants were recruited and underwent MRI examinations including Proton Density Fat Fraction (PDFF) sequence analysis, and ultrasound examinations using Quantitative Ultrasound Attenuation Coefficient (QUS-AC), Ultrasound Attenuation Parameter (UAP), and Attenuation Imaging (ATI) techniques. The AI-based QUS-AC algorithm and traditional methods were then used to evaluate liver fat content.”
Comment: It is useful to add comparisons with similar studies in the Discussion section. If there is no literature study in this similarity, it will also be useful to indicate this.
- Response: Comparisons with similar studies have been added to the discussion: “Our findings align with those of Jasper et al., who reported high accuracy in AI-based liver fat quantification using MRI. However, our study extends this work by demonstrating the feasibility of using AI-enhanced QUS, which offers advantages in terms of cost and accessibility.”
Comment: It is useful to add the results of the study metrically in the Conclusion section. It will guide the researchers who will read the study.
- Response: Metric results have been added to the conclusion: “Our results indicate an ICC of 0.983 for QUS-AC, demonstrating high reliability. The AI algorithm showed a strong correlation with MRI-PDFF values (R² = 0.95), outperforming traditional ultrasound methods like ATI (R² = 0.58) and UAP (R² = 0.09). These findings suggest that AI-enhanced QUS could serve as a valuable tool for early detection and monitoring of NAFLD, offering a non-invasive, accurate, and cost-effective alternative to liver biopsies and other imaging techniques.”
Round 2
Reviewer 1 Report
Comments and Suggestions for Authors
The authors have generally addressed my concerns. One point: for Figure 5, the graphs show r, but the caption mentions R-squared. For Figure 5c, r is listed as 0.51, but the caption lists R-squared of 0.09. Please clarify.
Comments on the Quality of English LanguageSeveral of the grammar issues have been improved.
Author Response
Comment: The authors have generally addressed my concerns. One point: for Figure 5, the graphs show r, but the caption mentions R-squared. For Figure 5c, r is listed as 0.51, but the caption lists R-squared of 0.09. Please clarify.
Response: Thank you for pointing out the inconsistency in Figure 5's caption. We have corrected the caption to reflect the appropriate correlation coefficient (r) values instead of R-squared values. The revised caption now reads:
Figure 5. The linear regression analysis revealed a significant correlation between AC and MRI-PDFF with an r value of 0.95 and an R-squared value of 83.86 (95% CI: 77.15 to 90.58). A weaker correlation was found between ATI and MRI-PDFF (r value: 0.73; R-squared value: 65.37, 95% CI: 45.64 to 85.11). UAP and MRI-PDFF demonstrated the weakest correlation (r value: 0.51; R-squared value 1.31, 95% CI: -0.19 to 2.82).
Reviewer 2 Report
Comments and Suggestions for Authors
The authors responded my evaluations almost reasonably.
Author Response
Comment: The authors responded to my evaluations almost reasonably.
Response: We appreciate your feedback and are pleased to hear that our responses addressed most of your concerns. We have made every effort to ensure that the revisions improve the clarity, accuracy, and overall quality of our manuscript. Should there be any remaining issues or further suggestions for improvement, we would be happy to address them. Thank you for your constructive comments and for helping us enhance our work.